# First Steps Toward Understanding the Extrapolation of Nonlinear Models to Unseen Domains

**Kefan Dong**
Stanford University
`kefandong@stanford.edu`

**Tengyu Ma**
Stanford University
`tengyuma@stanford.edu`

## Abstract

Real-world machine learning applications often involve deploying neural networks to domains that are not seen in the training time. Hence, we need to understand the extrapolation of *nonlinear* models—under what conditions on the distributions and function class, models can be guaranteed to extrapolate to new test distributions. The question is very challenging because even two-layer neural networks cannot be guaranteed to extrapolate outside the support of the training distribution without further assumptions on the domain shift. This paper makes some initial steps towards analyzing the extrapolation of nonlinear models for structured domain shift. We primarily consider settings where the *marginal* distribution of each coordinate of the data (or subset of coordinates) do not shift significantly across the training and test distributions, but the joint distribution may have a much bigger shift. We prove that the family of nonlinear models of the form $f(x) = \sum f_i(x_i)$, where $f_i$ is an *arbitrary* function on the subset of features $x_i$, can extrapolate to unseen distributions, if the covariance of the features is well-conditioned. To the best of our knowledge, this is the first result that goes beyond linear models and the bounded density ratio assumption, even though the assumptions on the distribution shift and function class are stylized.

## 1 Introduction

In real-world applications, machine learning models are often deployed on domains that are not seen in the training time. For example, we may train machine learning models for medical diagnosis on data from hospitals in Europe and then deploy them to hospitals in Asia.

Thus, we need to understand the extrapolation of models to new test distributions — how the model trained on one distribution behaves on another unseen distribution. This extrapolation of neural networks is central to various robustness questions such as domain generalization (Gulrajani & Lopez-Paz (2020); Ganin et al. (2016); Peters et al. (2016) and references therein) and adversarial robustness (Goodfellow et al., 2014; Kurakin et al., 2018), and also plays a critical role in nonlinear bandits and reinforcement learning where the distribution is constantly changing during training (Dong et al., 2021; Agarwal et al., 2019; Lattimore & Szepesvári, 2020; Sutton & Barto, 2018).

This paper focuses on the following mathematical abstraction of this extrapolation question:

*Under what conditions on the source distribution $P$, target distribution $Q$, and function class $\mathcal{F}$ do we have that any functions $f, g \in \mathcal{F}$ that agree on $P$ are also guaranteed to agree on $Q$?*

Here we can measure the agreement of two functions on $P$ by the $\ell_2$ distance between $f$ and $g$ under distribution $P$, that is, $\|f - g\|_P \triangleq \mathbb{E}_{x \sim P}[(f(x) - g(x))^2]^{1/2}$. The function $f$ can be thought of as the learned model, $g$ as the ground-truth function, and thus $\|f - g\|_P$ as the error on the source distribution $P$.

This question is well-understood for linear function class $\mathcal{F}$. Essentially, if the covariance of $Q$ can be bounded from above by the covariance of $P$ (in any direction), then the error on $Q$ is guaranteed

to be bounded by the error on $P$. We refer the reader to Lei et al. (2021); Mousavi Kalan et al. (2020) and references therein for more recent advances along this line.

By contrast, theoretical results for extrapolation of *nonlinear* models is rather limited. Classical results have long settled the case where $P$ and $Q$ have bounded density ratios (Ben-David & Urner, 2014; Sugiyama et al., 2007). Bounded density ratio implies that the support of $Q$ must be a subset of the support of $P$, and thus arguably these results do not capture the extrapolation behavior of models *outside* the training domain.

Without the bounded density ratio assumption, there was limited prior *positive* result for characterizing the extrapolation power of neural networks. Ben-David et al. (2010) show that the model can extrapolate when the $\mathcal{H}\Delta\mathcal{H}$-distance between training and test distribution is small. However, it remains unclear for what distributions and function class, the $\mathcal{H}\Delta\mathcal{H}$-distance can be bounded.[1] In general, the question is challenging partly because of the existence of such a strong impossibility result. As soon as the support of $Q$ is not contained in the support of $P$ (and they satisfy some non-degeneracy condition), it turns out that even two-layer neural networks cannot extrapolate—there are two-layer neural networks $f$ and $g$ that agree on $P$ perfectly but behave very differently on $Q$ (See Proposition 5 for a formal statement.)

The impossibility result suggests that any positive results on the extrapolation of nonlinear models require more fine-grained structures on the relationship between $P$ and $Q$ (which are common in practice Koh et al. (2021); Sagawa et al. (2022)) as well as the function class $\mathcal{F}$. The structure in the domain shift between $P$ and $Q$ may also need to be compatible with the assumption on the function class $\mathcal{F}$. This paper makes some first steps towards proving certain family of nonlinear models can extrapolate to a new test domain with structured shift.

We consider a setting where the joint distribution of the data can does not have much overlap across $P$ and $Q$ (and thus bounded density ratio assumption does not hold), whereas the marginal distributions for each coordinate of the data does overlap. Such a scenario may practically happen when the features (coordinates of the data) exhibit different correlations on the source and target distribution. For example, consider the task of predicting the probability of a lightning storm from basic meteorological information such as precipitation, temperature, etc. We learn models from some cities on the west coast of United States and deploy them to the east coast. In this case, the joint test distribution of the features may not necessarily have much overlap with the training distribution—correlation between precipitation and temperature could be vastly different across regions, e.g., the rainy season coincides with the winter's low temperature on the west coast, but not so much on the east coast. However, the individual feature's marginal distribution is much more likely to overlap between the source and target—the possible ranges of temperature on east and west coasts are similar.

Concretely, we assume that the features $x \in \mathbb{R}^{s_1+s_2}$ have Gaussian distributions and can be divided into two subsets $x_1 \in \mathbb{R}^{s_1}$ and $x_2 \in R^{s_2}$ such that each set of feature $x_i$ ($i \in \{1, 2\}$) has the same marginal distributions on $P$ and $Q$. Moreover, we assume that $x_1$ and $x_2$ are not exactly correlated on $P$—the covariance of features $x$ on distribution $P$ has a strictly positive minimum eigenvalue.

As argued before, restricted assumptions on the function class $\mathcal{F}$ are still necessary (for almost any $P$ and $Q$ without the bounded density ratio property). Here, we assume that $\mathcal{F}$ consists of all functions of the form $f(x) = f_1(x_1) + f_2(x_2)$ for *arbitrary* functions $f_1 : \mathcal{R}^{s_1} \to \mathbb{R}$ and $f_2 : \mathcal{R}^{s_2} \to \mathbb{R}$. The function class $\mathcal{F}$ does not contain all two-layer neural networks (so that the impossibility result does not apply), but still consists of a rich set of functions where each subset of features independently contribute to the prediction with arbitrary nonlinear transformations. We show that under these assumptions, if any two models approximately agree on $P$, they must also approximately agree on $Q$ — formally speaking, $\forall f, g \in \mathcal{F}, \|f - g\|_Q \lesssim \|f - g\|_P$ (Theorem 4).

We also prove a variant of the result above where we divide features vector $x \in \mathbb{R}^d$ into $d$ coordinate, denoted by $x = (x_1, \ldots, x_d)$ where $x_i \in \mathbb{R}$. The function class consists of all combinations of *nonlinear* transformations of $x_i$'s, that is, $\mathcal{F} = \{\sum_{i=1}^{d} f_i(x_i)\}$. Assuming coordinates of $x$ are pairwise Gaussian and a non-degenerate covariance matrix, the nonlinear model $f \in \mathcal{F}$ can extrapolate to any distribution $Q$ that has the same marginals as $P$ (Theorem 3).

---

[1]In fact, the $\mathcal{H}\Delta\mathcal{H}$-distance likely cannot be bounded when the function class contains two-layer neural networks, and the supports of the training and test distributions do not overlap —when there exists a function that can distinguish the source and target domain, the $\mathcal{H}\Delta\mathcal{H}$ divergence will be large.

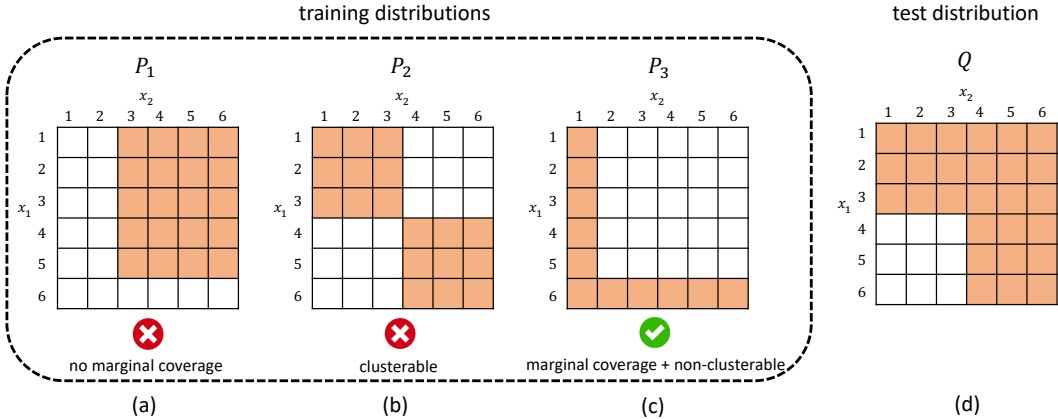

Figure 1: Visualization of three different training distributions $P_1, P_2, P_3$ and a test distribution $Q$, where the orange color blocks marks the support. **(a)** and **(b):** distributions $P_1, P_2$ that do not satisfy our conditions and cannot extrapolate. **(c):** a distribution $P_3$ that satisfies our conditions even though the support of $P_3$ is sparse. **(d):** the test distribution.

These results can be viewed as first steps for analyzing extrapolation beyond linear models. Compared the works of Lei et al. (2021) on linear models, our assumptions on the covariance of $P$ are qualitatively similar. We additionally require $P$ and $Q$ have overlapping marginal distributions because it is even necessary for the extrapolation of one-dimensional functions on a single coordinate. Our results work for a more expressive family of nonlinear functions, that is, $\mathcal{F} = \{\sum_{i=1}^{d} f_i(x_i)\}$, than the linear functions.

We also present a result on the case where $x_i$'s are discrete variables, which demonstrates the key intuition and also may be of its own interest. Suppose we have two discrete random variable $x_1$ and $x_2$. In this case, the joint distribution of $P$ and $Q$ can be both presented by a matrix (as visualized in Figure 1), and the marginal distributions are the column and row sums of this joint probability matrix. We prove that extrapolation occurs when (1) the support of the *marginals* of $Q$ is covered by $P$, and (2) the density matrix of $P$ is non-clusterable — we cannot find a subset of rows and columns such the support of $P$ in these rows and columns lies within their intersections.

In Figure 1, we visualize a few interesting cases. First, distributions $P_1, P_2$ visualized in Figures 1a and 1b, respectively, do not satisfy our conditions. Fundamentally, there are two models that agree on the support of distribution $P_1$ (or $P_2$), but still differ much on the non-support. In contrast, our result predict that models trained on distribution $P_3$ in Figure 1c can extrapolate to the distribution $Q$ in Figure 1d, despite that the support of $P_3$ is sparse and their support have very little overlap.

We also note that the failure of $P_1$ and $P_2$ demonstrate the non-triviality of our results. The overlapping marginal assumption by itself does not guarantees extrapolation, and condition (2), or analogously the minimal eigenvalue condition for the Gaussian cases, is critical for extrapolation.

Our proof technique for the theorems above is generally viewing $\|h\|_P^2$ as $K_P(h, h)$ for some kernel $K_P : \mathcal{F} \times \mathcal{F} \to \mathbb{R}$. Here $h$ is a shorthand for the error function $f - g$. Note that the kernel takes in two functions in $\mathcal{F}$ as inputs and captures relationship between the functions. Hence, the extrapolation of $\mathcal{F}$ (i.e., proving $\|h\|_Q \lesssim \|h\|_P$ for all $h \in \mathcal{F}$) reduces to the relationship of the kernels (i.e., whether $K_Q(h, h) \lesssim K_P(h, h)$ for all $h \in \mathcal{F}$), which is then governed by properties of the eigenspaces of kernel $K_P, K_Q$. Thanks to the special structure of our model class $\mathcal{F} = \sum f_i(x_i)$, we can analytically relate the eigenspace of the kernels $K_P, K_Q$ to more explicitly and interpretable property of the data distribution of $P$ and $Q$.

## 2 PROBLEM SETUP

We use $P$ and $Q$ to denote the source and target distribution over the space of features $\mathcal{X} = \mathcal{X}_1 \times \cdots \times \mathcal{X}_k$, respectively. We measure the extrapolation of a model class $\mathcal{F} \subseteq \mathbb{R}^{\mathcal{X}}$ from $P$ to $Q$

by the $\mathcal{F}$-restricted error ratio, or $\mathcal{F}$-RER as a shorthand:[2]

$$\tau(P, Q, \mathcal{F}) \triangleq \sup_{f,g \in \mathcal{F}} \frac{\mathbb{E}_Q[(f(x) - g(x))^2]}{\mathbb{E}_P[(f(x) - g(x))^2]}. \tag{1}$$

When $\tau(P, Q, \mathcal{F})$ is small, if two models $f, g \in \mathcal{F}$ approximately agree on $P$ (meaning that $\mathbb{E}_P[(f(x) - g(x))^2]$ is small), they must approximately agree on $Q$ because $\mathbb{E}_Q[(f(x) - g(x))^2] \leq \tau(P, Q, \mathcal{F})\mathbb{E}_P[(f(x) - g(x))^2]$.

The $\mathcal{F}$-restricted error ratio monotonically increases as we enlarge the model class $\mathcal{F}$, and eventually, $\tau(P, Q, \mathcal{F})$ becomes the density ratio between $Q$ and $P$ if the model $\mathcal{F}$ contains all functions with bounded output. To go beyond the bounded density ratio assumption, in this paper we focus on the structured model class $\mathcal{F} = \{\sum_{i=1}^k f_i(x_i) : \mathbb{E}_P[f_i(x_i)^2] < \infty, \forall i \in [k]\}$ where $f_i : \mathcal{X}_i \to \mathbb{R}$ is an *arbitrary* function. Since $\mathcal{F}$ is closed under addition, we can simplify Eq. (1) to $\tau(P, Q, \mathcal{F}) = \sup_{f \in \mathcal{F}} \frac{\mathbb{E}_Q[f(x)^2]}{\mathbb{E}_P[f(x)^2]}$. For simplicity, we omit the dependency on $P, Q, \mathcal{F}$ when the context is clear.

If the model class $\mathcal{F}$ includes the ground-truth labeling function, $\tau(P, Q, \mathcal{F})$ upperbounds the ratio between the error on distribution $Q$ and the error on distribution $P$ (formally stated in Proposition 1), which provides the robustness guarantee of the trained model. This is because when $g$ corresponds to the ground-truth label, $\mathbb{E}_P[(f(x) - g(x))^2]$ becomes the $\ell_2$ error of model $f$.

**Proposition 1.** *Let $\tau$ be the $\mathcal{F}$-RER defined in Eq.* (1). *For any distribution $P, Q$ and model class $\mathcal{F}$, if there exists a model $f^\star$ in $\mathcal{F}$ that can represent the true labeling $y : \mathcal{X} \to \mathbb{R}$ on both $P$ and $Q$:*

$$\mathbb{E}_{\frac{1}{2}(P+Q)}[(y(x) - f^\star(x))^2] \leq \epsilon_{\mathcal{F}}, \tag{2}$$

*then we have*

$$\forall f \in \mathcal{F}, \quad \mathbb{E}_Q[(y(x) - f(x))^2] \leq (8\tau + 4)\epsilon_{\mathcal{F}} + 4\tau\mathbb{E}_P[(y(x) - f(x))^2]. \tag{3}$$

Proof of this proposition is deferred to Appendix A.1

**Relationship to the $\mathcal{H}\Delta\mathcal{H}$-distance.** Compared with the $\mathcal{H}\Delta\mathcal{H}$-distance (Ben-David et al., 2010):

$$d_{\mathcal{H}\Delta\mathcal{H}}(P, Q) = 2\sup_{f,g \in \mathcal{F}} |\mathrm{Pr}_{x \sim P}[f(x) \neq g(x)] - \mathrm{Pr}_{x \sim Q}[f(x) \neq g(x)]|, \tag{4}$$

our differences are: (1) we consider $\ell_2$ loss instead of classification loss, and (2) $\tau$ focuses on the ratio of losses whereas $d_{\mathcal{H}\Delta\mathcal{H}}$ focuses on the absolute difference. As we will see later, these differences bring the mathematical simplicity to prove concrete conditions for model extrapolation.

**Additional notations.** Let $\mathbb{I}[E]$ be the indicator function that equals 1 if the condition $E$ is true, and 0 otherwise. For an integer $n$, let $[n]$ be the set $\{1, 2, \cdots, n\}$. For a vector $x \in \mathbb{R}^d$, we use $[x]_i$ to denote its $i$-th coordinate. Similarly, $[M]_{i,j}$ denotes the $(i, j)$-th element of a matrix $M$. We use $M^{\odot n}$ to represent the element-wise $n$-th power of the matrix $M$ (i.e., $[M^{\odot n}]_{i,j} = ([M]_{i,j})^n$). Let $I_d \in \mathbb{R}^{d \times d}$ be the identity matrix, $\mathbf{1}_d \in \mathbb{R}^d$ the all-1 vector and $e_{i,d}$ the $i$-th base vector. We omit the subscript $d$ when the context is clear. For a square matrix $P \in \mathbb{R}^{d \times d}$, we use $\mathrm{diag}(P) \in \mathbb{R}^{d \times d}$ to denote the matrix generated by masking out all non-diagonal terms of $P$. For list $\sigma_1, \cdots, \sigma_d$, let $\mathrm{diag}(\{\sigma_1, \cdots, \sigma_d\}) \in \mathbb{R}^{d \times d}$ be the diagonal matrix whose diagonal terms are $\sigma_1, \cdots, \sigma_d$.

For a symmetric matrix $M \in \mathbb{R}^{d \times d}$, let $\lambda_1(M) \leq \lambda_2(M) \leq \cdots \leq \lambda_d(M)$ be its eigenvalues in ascending order, and $\lambda_{\max}(M), \lambda_{\min}(M)$ the maximum and minimum eigenvalue, respectively. Similarly, we use $\sigma_1(M), \cdots, \sigma_{\min(d_1, d_2)}(M)$ to denote the singular values of $M \in \mathbb{R}^{d_1 \times d_2}$.

## 3 MAIN RESULTS

In this section, we present our main results. Section 3.1 discusses the case where the features have discrete values. In Section 3.2 and 3.3, we extend our analysis to two other settings with real-valued features.

---

[2]For simplicity, we set $0/0 = 0$.

### 3.1 FEATURES WITH DISCRETE VALUES

For better exposition, we discuss the case that $x = (x_1, x_2)$ here and defer the discussion of the general case to Appendix A.5. We assume that $x_i$ takes the value in $\{1, 2, \cdots, r_i\}$ for $i \in [2]$. Hence, the density of distribution $P$ can be written in a matrix with dimension $r_1 \times r_2$.

We measure the (approximate) clusterablity by eigenvalues of the Laplacian matrix of a bipartite graph associated with the density matrix $P \in \mathbb{R}^{r_1 \times r_2}$, which is known to capture the clustering structure of the graph (Chung, 1996; Alon, 1986). Let $G_P$ be a weighted bipartite graph whose adjacency matrix equals the density matrix $P \in \mathbb{R}^{r_1 \times r_2}$—concretely, $U = \{u_1, \cdots, u_{r_1}\}$ and $V = \{v_1, \cdots, v_{r_2}\}$ are the sets of vertices, and the weight between $u_i, v_j$ is $P(x_1 = i, x_2 = j)$. To define the signless Laplacian of $G_P$, let $d_1 \in \mathbb{R}^{r_1}$ and $d_2 \in \mathbb{R}^{r_2}$ be the row and column sums of the weight matrix $P$ (in other words, degree of the vertices), and $D_1 = \text{diag}(d_1) \in \mathbb{R}^{r_1 \times r_1}$, $D_2 = \text{diag}(d_2) \in \mathbb{R}^{r_2 \times r_2}$ the diagonal matrices induced by $d_1, d_2$, respectively. The signless Laplacian $K_P$ and normalized signless Laplacian $\bar{K}_P$ are:

$$K_P = \begin{pmatrix} D_1 & P \\ P^\top & D_2 \end{pmatrix}, \quad \bar{K}_P = \text{diag}(K_P)^{-1/2} K_P \text{diag}(K_P)^{-1/2}. \tag{5}$$

Compared with the standard Laplacian matrix, the non-diagonal terms in the signless Laplacian $K_P$ are positive and equal to the absolute value of corresponding terms in the standard Laplacian matrix. In the following theorem, we bound the $\mathcal{F}$-RER from above by the eigenvalues of $\bar{K}_P$ and the density ratio of the marginal distributions of $Q, P$.

**Theorem 2.** *For any distributions $P, Q$ over discrete random variables $x_1, x_2$, and the model class $\mathcal{F} = \{f_1(x_1) + f_2(x_2)\}$ where $f_i : \mathcal{X}_i \to \mathbb{R}$ is an arbitrary function, the $\mathcal{F}$-RER can be bounded above:*

$$\tau(P, Q, \mathcal{F}) \leq 2\lambda_2(\bar{K}_P)^{-1} \max_{i \in [2], t \in [r_i]} \frac{Q(x_i = t)}{P(x_i = t)}. \tag{6}$$

Compared with prior works that assumes a bounded density ratio on the entire distribution (e.g., Ben-David & Urner (2014); Sugiyama et al. (2007)), we only require a bounded density ratio of the marginal distributions. In other words, the model class $f(x) = f_1(x_1) + f_2(x_2)$ can extrapolate to distributions with a larger support (see Figure 1c). In contrast, for an unstructured model (i.e., $f(x)$ is an arbitrary function of the entire input $x$), the model can behave arbitrarily on data points outside the support of $P$.

Qualitatively, Theorem 2 proves sufficient conditions under which the structured model class can extrapolate to unseen distributions (as visualized in Figure 1)—In particulary, Theorem 2 implies that for non-trivial extrapolation, that is, $\tau(P, Q, \mathcal{F}) < \infty$, we need (a) $\max_{i \in [2], t \in [r_i]} \frac{Q(x_i = t)}{P(x_i = t)} < \infty$ (i.e., the support of the *marginals* of $Q$ is covered by $P$), and (b) $\lambda_2(\bar{K}_P) > 0$. To interpret condition (b), note that Cheeger's inequality implies that $\lambda_2(\bar{K}_P) > 0$ if and only if the bipartite graph $G_P$ is connected (Chung, 1996; Alon, 1986)[3], that is, there does not exist non-empty strict subsets of vertices $U' \subset U, V' \subset V$, such that $P(x_1 \in U', x_2 \notin V') = 0$ and $P(x_1 \notin U', x_2 \in V') = 0$. Equivalent, we cannot shuffle the rows and columns of $P$ to form a block diagonal matrix where each block is a strict sub-matrix of $P$. In other words, the density matrix $P$ is non-clusterable as discussed in Section 1.

**Proof sketch of Theorem 2.** In the following we present a proof sketch of Theorem 2. We start with a high-level proof strategy and then instantiate the proof on the setting of Theorem 2.

Suppose we can find a set of (not necessarily orthogonal) basis $\{b_1, \cdots, b_r\}$ where $b_i : \mathcal{X} \to \mathbb{R}$, such that any model $f \in \mathcal{F}$ can be represented as a linear combination of basis, that is, $f = \sum_{i=1}^r v_i b_i$. Since the model family $\mathcal{F}$ is closed under subtraction, we have

$$\tau(P, Q, \mathcal{F}) = \sup_{f \in \mathcal{F}} \frac{\|f\|_Q^2}{\|f\|_P^2} = \sup_{v \in \mathbb{R}^r} \frac{\mathbb{E}_Q\left[\left(\sum_{i=1}^r v_i b_i(x)\right)^2\right]}{\mathbb{E}_P\left[\left(\sum_{i=1}^r v_i b_i(x)\right)^2\right]} = \sup_{v \in \mathbb{R}^r} \frac{\sum_{i,j=1}^r [v]_i [v]_j \mathbb{E}_Q[b_i(x) b_j(x)]}{\sum_{i,j=1}^r [v]_i [v]_j \mathbb{E}_P[b_i(x) b_j(x)]}. \tag{7}$$

---

[3]Cheeger's inequality measures the clustering structure of a graph by the eigenvalues of its standard Laplacian. However, the signless Laplacian and standard Laplacian have the same eigenvalues for bipartite graphs Cvetković et al. (2007); Grone et al. (1990).

If we define the kernel matrices $[K_P]_{i,j} = \mathbb{E}_P[b_i(x)b_j(x)]$ and $[K_Q]_{i,j} = \mathbb{E}_Q[b_i(x)b_j(x)]$ (we use the same notation for the kernel matrix and signless Laplacian because later we will show that the kernels $K_P, K_Q$ equal to the signless Laplacian of the bipartite graphs $G_P, G_Q$ with specific choice of the basis $b_i$) , it follows that

$$\sup_{v \in \mathbb{R}^r} \frac{\sum_{i,j=1}^r v_i v_j \mathbb{E}_Q[b_i(x)b_j(x)]}{\sum_{i,j=1}^r v_i v_j \mathbb{E}_P[b_i(x)b_j(x)]} = \sup_{v \in \mathbb{R}^r} \frac{v^\top K_Q v}{v^\top K_P v}. \tag{8}$$

Hence, upper bounding $\tau(P, Q, \mathcal{F})$ reduces to bounding the eigenvalues of kernel matrices $K_P, K_Q$.

Since the model has the structure $f(x) = f_1(x_1) + f_2(x_2)$, we can construct the basis $\{b_t\}_{t=1}^r$ explicitly. For any $i \in [2], t \in [r_i]$, with little abuse of notation, let $b_{i,t}(x) = \mathbb{I}[x_i = t]$. We can verify that the set $\{b_{i,t}\}_{i \in [2], t \in [r_i]}$ is indeed a complete set of basis. As a result, the kernel matrices $K_P$ can be computed directly using its definition:

$$\mathbb{E}_P[b_{i,t}(x)b_{j,s}(x)] = \begin{cases} P(x_i = t, x_j = s), & \text{when } i \neq j, \\ P(x_i = t)\mathbb{I}[s = t], & \text{when } i = j, \end{cases} \tag{9}$$

which is exactly the Laplacian matrix defined in Eq. (5).

To prove Eq. (6), we need to upperbound the eigenvalues of $K_Q$. Since the eigenvalues of the normalized signless Laplacian $\bar{K}_Q$ is universally upper bounded by 2 for every distribution $Q$, we first write $K_P, K_Q$ in terms of $\bar{K}_P, \bar{K}_Q$. Formally, let $D_P = \mathrm{diag}(K_P)$ and $D_Q = \mathrm{diag}(K_Q)$ and we have

$$\sup_{v \in \mathbb{R}^r} \frac{v^\top K_Q v}{v^\top K_P v} = \sup_{v \in \mathbb{R}^r} \frac{v^\top D_Q^{1/2} \bar{K}_Q D_Q^{1/2} v}{v^\top D_P^{1/2} \bar{K}_P D_P^{1/2} v} \leq \frac{\lambda_{\max}(\bar{K}_Q)}{\lambda_{\min}(\bar{K}_P)} \sup_{v \in \mathbb{R}^r} \frac{\|D_Q^{1/2} v\|_2^2}{\|D_P^{1/2} v\|_2^2}. \tag{10}$$

However, this naive bound is vacuous because for any $P$ we have $\lambda_{\min}(\bar{K}_P) = 0$. In fact, $K_P$ and $K_Q$ share the eigenvalue 0 and the corresponding eigenvector $u \in \mathbb{R}^{r_1 + r_2}$ with $[u]_t = (-1)^{\mathbb{I}[t > r_1]}$. Therefore we can restrict to the subspace orthogonal to the direction $u$, and then $\lambda_{\min}(\bar{K}_P)$ becomes $\lambda_2(\bar{K}_P)$ in Eq. (10). Finally, by basic algebra we also have $\lambda_{\max}(\bar{K}_Q) \leq 2$ and $\sup_{v \in \mathbb{R}^r} \frac{\|D_Q^{1/2} v\|_2^2}{\|D_P^{1/2} v\|_2^2} \leq \max_{i \in [2], t \in [r_i]} \frac{Q(x_i = t)}{P(x_i = t)}$, which complete the proof sketch. The full proof of Theorem 2 is deferred to Appendix A.3.

## 3.2 FEATURES WITH REAL VALUES

In this section we extend our analysis to the case where $x_1, x_2, \cdots, x_d$ are real-valued random variables. Recall that our model has the structure $f(x) = \sum_{i=1}^d f_i(x_i)$ where $f_i$ is an arbitrary one-dimensional function.

When $d = 2$, we can view this setting as a direct extension of the setting in Section 3.1 where each $x_i$'s has infinite number of possible values (instead of finite number), and thus the Laplacian "matrix" becomes infinite-dimensional. Nonetheless, we can still bound the $\mathcal{F}$-RER from above, as stated in the following theorem.

**Theorem 3.** *For any distributions $P, Q$ over variables $x = (x_1, \cdots, x_d)$ with matching marginals, assume that $(x_i, x_j)$ has the distribution of a two-dimensional Gaussian random variable for every $i, j \in [d]$. Let $\tilde{x} = (\tilde{x}_1, \cdots, \tilde{x}_d)$ be the normalized input where $\tilde{x}_i \triangleq (x_i - \mathbb{E}_P[x_i])\mathrm{Var}(x_i)^{-1/2}$ has zero mean and unit variance for every $i \in [d]$, and $\tilde{\Sigma}_P \triangleq \mathbb{E}_P[\tilde{x}\tilde{x}^\top]$ the covariance matrix of $\tilde{x}$. Then*

$$\tau(P, Q, \mathcal{F}) \leq \frac{d}{\lambda_{\min}(\tilde{\Sigma}_P)}. \tag{11}$$

For better exposition, we first focus on the case where every $x_i$ has zero mean and unit variance, hence $\tilde{x} = x$ and $\tilde{\Sigma}_P = \Sigma_P \triangleq \mathbb{E}_P[xx^\top]$. Compared with linear models, Theorem 3 proves that the structured nonlinear model class $f(x) = \sum_{i=1}^d f_i(x_i)$ can extrapolate with similar conditions—for linear models $\mathcal{F}_{\mathrm{linear}} \triangleq \{v^\top x : v \in \mathbb{R}^d\}$ we have

$$\tau(P, Q, \mathcal{F}_{\mathrm{linear}}) = \sup_{v \in \mathbb{R}^d} \frac{\|v^\top x\|_Q^2}{\|v^\top x\|_P^2} = \sup_{v \in \mathbb{R}^d} \frac{v^\top \mathbb{E}_Q[xx^\top]v}{v^\top \mathbb{E}_P[xx^\top]v} \lesssim \lambda_{\min}(\Sigma_P)^{-1} = \lambda_{\min}(\tilde{\Sigma}_P)^{-1},$$

which equals to the RHS of Eq. (11) up to factors that does not depend on the covariance $\Sigma_P, \Sigma_Q$.

We emphasize that we only assume the marginals on every *pair* of features $x_i, x_j$ is Gaussian, which does not imply the Gaussianity of the joint distribution of $x$. In fact, there exists a non-Gaussian distribution that satisfies our assumption.

**Proof sketch of Theorem 3.** On a high level, we treat the features $x_i$'s as discrete random variables, and follow the same proof strategy as in Theorem 2. For better exposition, we first assume that $x_i$ has zero mean and unit variance for every $i \in [d]$, hence $\tilde{\Sigma}_P = \Sigma_P \triangleq \mathbb{E}_P[xx\top]$.

First we consider a simplified case when $d = 2$. Because $x_1, x_2$ are continuous, the normalized signless Laplacian $\bar{K}_P$ is infinite dimensional, and has the form $\bar{K}_P = \begin{pmatrix} I & A \\ A^\top & I \end{pmatrix}$, where $A$ is an infinite dimensional "matrix" indexed by real numbers $x_1, x_2 \in \mathbb{R}$ with values $[A]_{x_1,x_2} = P(x_1, x_2)/\sqrt{P(x_1)P(x_2)}$, and $I$ is the identity "matrix". Recall the proof of Theorem 2 gives

$$\tau(P, Q, \mathcal{F}) \le 2\lambda_2(\bar{K}_P)^{-1} \max_{i \in [2], t} \frac{Q(x_i = t)}{P(x_i = t)}. \tag{12}$$

By the assumption that $P, Q$ have matching marginals, we get $\max_{i \in [2], t} \frac{Q(x_i=t)}{P(x_i=t)} = 1$. As result, we only need to lowerbound the second smallest eigenvalue of $\bar{K}_P$. To this end, we first write $A$ in its singular value decomposition form $A = U\Lambda V^\top$, where $UU^\top = I, VV^\top = I$ and $\Lambda = \text{diag}(\{\sigma_n\}_{n \ge 0})$ with $\sigma_0 \ge \sigma_1 \ge \cdots$. Then we get

$$\bar{K}_P = \begin{pmatrix} I & A \\ A^\top & I \end{pmatrix} = \begin{pmatrix} U & 0 \\ 0 & V \end{pmatrix} \begin{pmatrix} I & \Lambda \\ \Lambda^\top & I \end{pmatrix} \begin{pmatrix} U^\top & 0 \\ 0 & V^\top \end{pmatrix}. \tag{13}$$

Since the matrix $\hat{K}_P \triangleq \begin{pmatrix} I & \Lambda \\ \Lambda^\top & I \end{pmatrix}$ consists of four diagonal sub-matrices, we can shuffle the rows and columns of $\hat{K}_P$ to form a block-diagonal matrix with blocks $\left\{ \begin{pmatrix} 1 & \sigma_n \\ \sigma_n & 1 \end{pmatrix} \right\}_{n=0,1,2,\cdots}$. As a result, the eigenvalues of $\hat{K}_P$ are $1 \pm \sigma_0, 1 \pm \sigma_1, \cdots$. Because $1 = \sigma_0 \ge \sigma_1 \ge \cdots \ge 0$, the smallest and second smallest eigenvalues of $\hat{K}_P$ are $1 - \sigma_0$ and $1 - \sigma_1$, respectively, meaning that $\lambda_2(\bar{K}_P) = \lambda_2(\hat{K}_P) = 1 - \sigma_1$. By the assumption that $(x_1, x_2)$ follows from Gaussian distribution, the "matrix" $A$ is a Gaussian kernel, whose eigenvalues and eigenfunctions can be computed analytically— Theorem 11 proves that $\sigma_1 = |\mathbb{E}_P[x_1 x_2]|$ if $x_1, x_2$ have zero mean and unit variance. Consequently, $\lambda_2(\bar{K}_P) = 1 - \sigma_1 = 1 - |\mathbb{E}_P[x_1 x_2]| = \lambda_{\min}(\Sigma_P)$.

Now we briefly discuss the case when $d = 3$, and the most general cases (i.e., $d > 3$) are proved similarly. When $d = 3$, the normalized kernel will have the following form

$$\bar{K}_P = \begin{pmatrix} I & A & B \\ A^\top & I & C \\ B^\top & C^\top & I \end{pmatrix}. \tag{14}$$

By the assumption that $x_1, x_2$ are zero mean and unit variance with joint Gaussian distribution, matrices $A$ is symmetric because $[A]_{x_1,x_2} = P(x_1,x_2)/\sqrt{P(x_1)P(x_2)} = P(x_2,x_1)/\sqrt{P(x_1)P(x_2)} = [A]_{x_2,x_1}$ Similarly, matrices $B, C$ are symmetric. In addition, Theorem 11 shows that the eigenfunctions of the Gaussian kernel is *independent* of the value $\mathbb{E}_P[x_i x_j]$. Hence, the matrices $A, B, C$ shares the same eigenspace and can be diagonalized simultaneously:

$$\bar{K}_P = \begin{pmatrix} I & A & B \\ A^\top & I & C \\ B^\top & C^\top & I \end{pmatrix} = \begin{pmatrix} U & 0 & 0 \\ 0 & U & 0 \\ 0 & 0 & U \end{pmatrix} \begin{pmatrix} I & \Lambda_A & \Lambda_B \\ \Lambda_A^\top & I & \Lambda_C \\ \Lambda_B^\top & \Lambda_C^\top & I \end{pmatrix} \begin{pmatrix} U^\top & 0 & 0 \\ 0 & U^\top & 0 \\ 0 & 0 & U^\top \end{pmatrix}. \tag{15}$$

By reshuffling the columns and rows, the eigenvalues of $\bar{K}_P$ are the union of the eigenvalues of following matrices

$$\{\hat{K}_P^{(n)}\}_{n=0,1,2,\cdots} \triangleq \left\{ \begin{pmatrix} 1 & \sigma_n(A) & \sigma_n(B) \\ \sigma_n(A) & 1 & \sigma_n(C) \\ \sigma_n(B) & \sigma_n(C) & 1 \end{pmatrix} \right\}_{n=0,1,2,\cdots}. \tag{16}$$

Theorem 11 implies that $\sigma_n(A) = ([\Sigma_P]_{1,2})^n, \sigma_n(B) = ([\Sigma_P]_{1,3})^n$ and $\sigma_n(C) = ([\Sigma_P]_{2,3})^n$. Consequently we get $\hat{K}_P^{(n)} = \Sigma_P^{\odot n}$. Then, this theorem follows directly by noticing $\lambda_{\min}(\Sigma_P^{\odot n}) \geq \lambda_{\min}(\Sigma_P)$ for $n \geq 1$ (Lemma 13).

Finally, the general case where $x_i$ has arbitrary mean and variance can be reduced to the case where $x_i$ has zero mean and unit variance (Lemma 8). The full proof of Theorem 11 is deferred to Appendix A.6.

### 3.3 TWO FEATURES WITH MULTI-DIMENSIONAL GAUSSIAN DISTRIBUTION

Now we extend Theorem 3 to the case where $x_1 \in \mathbb{R}^{d_1}, x_2 \in \mathbb{R}^{d_2}$ are two subset of features with dimensions $d_1, d_2 > 1$, respectively, and the input $x = (x_1, x_2)$ has Gaussian distribution. Recall that the model class is $\mathcal{F} = \{f_1(x_1) + f_2(x_2) : \mathbb{E}_P[f_i(x_i)^2] < \infty, \forall i \in [2]\}$. In this case, we can still upper bound the $\mathcal{F}$-RER by the eigenvalues of the covariance matrix $\Sigma_P$, which is stated in the following theorem.

**Theorem 4.** *For any distributions $P, Q$ over variables $x = (x_1, x_2)$ where $x_1 \in \mathbb{R}^{d_1}, x_2 \in \mathbb{R}^{d_2}$, let $\Sigma_P = \mathbb{E}_P[xx^\top]$. If $x = (x_1, x_2)$ has Gaussian distribution on both $P$ and $Q$ with zero mean and matching marginals, and $\mathbb{E}_P[x_1 x_1^\top] = I, \mathbb{E}_P[x_2 x_2^\top] = I$, then*

$$\tau(P, Q, \mathcal{F}) \leq \frac{2}{\lambda_{\min}(\Sigma_P)}. \tag{17}$$

We defer the proof of Theorem 4 to Appendix A.7.

Compared with Theorem 3 where the features $x_1, \cdots, x_d$ are one-dimensional, our condition for the covariance is almost the same: $\lambda_{\min}(\Sigma_P) > 0$. However, the model class considered Theorem 4 is more powerful because it captures nonlinear interactions between features within the same subset. As a compromise, the assumption on the marginals of $P$ and $Q$ is stronger because Theorem 4 requires matching marginals on each subset of the features, whereas Theorem 3 only requires matching marginals on each individual feature.

**Remarks.** Our current techniques can only handle the case when the input is divided into $k = 2$ subsets. This is because for $k \geq 3$ we must diagonalize multiple multi-dimensional Gaussian kernels simultaneously using the same set of eigenfunctions, as required in the proof of Theorem 3. However, these multi-dimensional Gaussian kernels do not share the same eigenfunctions because the rotation matrix $U, V$ depends on the covariance $\mathbb{E}_P[x_i x_j^\top]$. Hence, the proof strategy for Theorem 3 fails for the case $k \geq 3$.

## 4 LOWER BOUNDS

In this section, we prove a lower bound as a motivation to consider structured distributions shifts. The following proposition shows that in the worst case, models learned on $P$ cannot extrapolate to $Q$ when the support of distribution $Q$ is not contained in the support of $P$.

**Proposition 5.** *Let the model class $\mathcal{F}$ be the family of two-layer neural networks with ReLU activation: $\mathcal{F} = \left\{\sum_i a_i \mathrm{ReLU}(w_i^\top x + b_i) : w_i \in \mathbb{R}^d, a_i, b_i \in \mathbb{R}\right\}$. Suppose for simplicity that all the inputs have unit norm (i.e., $\|x\|_2 = 1$). If $Q$ has non-zero probability mass on the set of points well-separated from the support of $P$ in the sense that*

$$\exists \epsilon > 0, \quad Q(\{x : \|x\|_2 = 1, \mathrm{dist}(x, \mathrm{supp}(P)) \geq \epsilon\}) > 0, \tag{18}$$

*we can construct a model $f \in \mathcal{F}$ such that $\|f\|_P = 0$ but $\|f\|_Q$ can be arbitrarily large.*

A complete proof of this proposition is deferred to Appendix A.8. On a high level, we prove this proposition by construct a two-layer neural network $g_t$ that represents a bump function around any given input $t \in S^{d-1}$. As a result, when $t$ is a point in $\mathrm{supp}(Q) \setminus \mathrm{supp}(P)$, the model $g_t(x)$ will have zero $\ell_2$ norm on $P$ but have a positive $\ell_2$ norm on $Q$. This construction is inspired by Dong et al. (2021, Theorem 5.1).

## 5 RELATED WORKS

The most related work is Ben-David et al. (2010), where they use the $\mathcal{H}\Delta\mathcal{H}$-distance to measure the maximum discrepancy of two models $f, g \in \mathcal{F}$ on any distributions $P, Q$. However, it remains an open question to determine when $\mathcal{H}\Delta\mathcal{H}$-distance is small for *concrete* nonlinear model classes and distributions. On the technical side, the quantity $\tau$ is an analog of the $\mathcal{H}\Delta\mathcal{H}$-distance for regression problems, and we provide concrete examples where $\tau$ is upper bounded even if the distributions $P, Q$ have significantly different support.

Another closely related settings are domain adaptation (Ganin & Lempitsky, 2015; Ghifary et al., 2016; Ganin et al., 2016) and domain generalization (Gulrajani & Lopez-Paz, 2020; Peters et al., 2016), where the algorithms actively improve the extrapolation of learned model either by using unlabeled data from the test domain (Sun & Saenko, 2016; Li et al., 2020a;b; Zhang et al., 2019), or learn an invariant model across different domains (Arjovsky et al., 2019; Peters et al., 2016; Gulrajani & Lopez-Paz, 2020). There are also algorithms that learn features whose distributions on the source and target domain have a small discrepancy measured by the maximum mean discrepancy (Donahue et al., 2014; Long et al., 2015), or the Wasserstein distances (Shen et al., 2018; Courty et al., 2017). In comparison, this paper studies whether a model trained on one distribution (without any implicit bias and unlabeled data from test domain) extrapolates to new distributions. There are also prior works that theoretically analyze algorithms that use additional (unlabeled) data from the test distribution, such as self-training (Wei et al., 2020; Chen et al., 2020), contrastive learning (Shen et al., 2022; HaoChen et al., 2022), label propagation (Cai et al., 2021), etc.

## 6 CONCLUSIONS

In this paper, we propose to study domain shifts between $P$ and $Q$ with the structure that each feature's marginal distribution has good overlap between source and target domain but the joint distribution of the features may have a much bigger shift. As a first step toward understanding the extrapolation of nonlinear models, we prove sufficient conditions for the model $f(x) = \sum_{i=1}^{k} f_i(x_i)$ to extrapolate where $f_i$ is an arbitrary function of a single feature.

Even though the assumptions on the shift and function class is stylized, to the best of our knowledge, this is the first analysis of how nonlinear models extrapolate when source and target distribution *do not* have shared support in concrete settings. There still remain many interesting open questions, which we leave as future works:

(a) Our current proof can only deal with a restricted nonlinear model family of the special form $f(x) = \sum_{i=1}^{k} f_i(x_i)$. Can we extend to a more general model class?
(b) In this paper, we focus on regression tasks with $\ell_2$ loss for mathematical simplicity, whereas majority of the prior works focus on the classification problems. Do similar results also hold for classification problem?

ACKNOWLEDGMENT

The authors would like to thank Yuanhao Wang, Yuhao Zhou, Hong Liu, Ananya Kumar, Jason D. Lee, and Kendrick Shen for helpful discussions. The authors would also like to thank the support from NSF CIF 2212263.

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
