# OpenReview forum: "First Steps Toward Understanding the Extrapolation of Nonlinear Models to Unseen Domains"
_ICLR.cc/2023/Conference — ICLR 2023 poster_

### Official Review · Reviewer_1mvE · 2022-10-24

**Confidence:** 3
**Correctness:** 4
**Technical Novelty And Significance:** 4
**Empirical Novelty And Significance:** Not applicable
**Recommendation:** 6

**Clarity, Quality, Novelty And Reproducibility:**

The work is largely clear, well-written, and easy to follow. The results build off existing work but generalize it to a new and novel setting that is very relevant to the field.

**Strength And Weaknesses:**

The paper addresses a central and important question in the theory of generalization, and extends new results to a broad class of nonlinear functions without making assumptions about the support that fail in the context of domain generalization. Their arguments are rigorous and well-presented, and make a convincing case for why this result is powerful and novel. Their argument convincingly shows how the ability to extrapolate to new domains (for the class of functions they consider) is tied to the conditions of non-degenerate covariance and overlapping marginal support.

My primary hesitation with the paper is in the assumptions made about the data. My understanding is that the results shown in this paper hold for three settings: 1) when x takes on discrete values, 2) when $x=(x_1,...,x_d)$ such that each pair $(x_i, x_j)$ is pairwise gaussian distributed, and 3) when x can be divided into two subsets $x_1,x_2$ and $(x_1,x_2)$ is gaussian distributed.

First, the results in the discrete setting are proved only for the 2-dimensional setting as far as I can tell. In the paper, it is stated that "Theorem 2 can be easily extend to the case when k > 2. We choose to present the current version because of simplicity, and because the kernel matrix KP coincides with the Laplacian of a bipartite graph when k = 2, which provides further insights and better intuitions." I think the extension to higher dimensional x should at least be shown in the appendices.

Second, the assumptions on the data in real-valued settings seem quite restrictive. I am not an expert on learning theory, so it is possible these assumptions are more standard than I assume, but the first set of assumptions (e.g. each pair of features is pairwise gaussian distributed) seems unlikely to be true in general cases, and the note in the latter case (where features are partitioned into subsets) that "we want to handle the general version where the inputs are divided into multiple subsets, but due to technical reasons we can only use our proof techniques to two subsets" also seems like a significant constraint. Perhaps the authors could give some motivation for whether there is a reason to expect either of these assumptions about the form of the data to hold in cases that are of particular practical or theoretical interest.

**Summary Of The Paper:**

This paper addresses the problem of understanding how models generalize to new domains from a theoretical perspective. Specifically, they seek to answer the core question:
"Under what conditions on the source distribution P , target distribution Q, and function class F do we have that any functions f, g ∈ F that agree on P are also guaranteed to agree on Q?"
This paper extends existing results on this problem to a broad class of nonlinear functions, without requiring the assumption that the support of Q lies within the support of P. They show a number of related results for different types of data (discrete, continuous gaussian-distributed) relying on the assumptions that a) the support of the *marginals* of Q lie within the support of the marginals of P, and b) the covariance-matrix is non-degenerate, then the function class $F={\sum_i f_i(x_i)}$ along with P and Q satisfy the above property.

**Summary Of The Review:**

This paper tackles a very interesting theoretical question and attempts to provide answers in a setting that is a) of direct practical interest and b) has not yet been satisfactorily resolved in the literature. It is well-written and high-quality, but the specific assumptions made about the form of the data may be overly restrictive and limit its usefulness. For now I would consider this paper borderline, but I hope that the authors or other reviewers might be able to provide more context on how limiting those assumptions really are. From an empirical perspective they seem quite severe, but from the perspective of learning theory it may still be a significant result, and as such I may update my score pending further discussion.

---

> ### Author Response · Authors · 2022-11-11
> **Response to Reviewer 1mvE**
>
> We thank the reviewer for the comments, and for noting our results “make a convincing case for why this result is powerful and novel”. In the following, we will address the reviewer’s comments in detail.
>
> > the results in the discrete setting are proved only for the 2-dimensional setting as far as I can tell … I think the extension to higher dimensional x should at least be shown in the appendices.
>
> We thank the reviewer for their suggestions. We’ve updated the paper to include the proof for $k>2$ (Theorem 8 on Page 14)
>
> > the first set of assumptions (e.g. each pair of features is pairwise gaussian distributed) seems unlikely to be true in general cases … Perhaps the authors could give some motivation for whether there is a reason to expect either of these assumptions about the form of the data to hold in cases that are of particular practical or theoretical interest.
>
> We kindly refer the reviewer to the separate reply for our response to this question.
>
> > in the latter case (where features are partitioned into subsets) that "we want to handle the general version where the inputs are divided into multiple subsets, but due to technical reasons we can only use our proof techniques to two subsets" also seems like a significant constraint.
>
> We agree with the reviewer that this is a limitation of our paper. However, we’d still like to emphasize that, even though the assumptions are stylized, our paper proves the *first* positive results for the extrapolation of nonlinear models in concrete settings.

---

### Official Review · Reviewer_JEBk · 2022-10-26

**Confidence:** 3
**Correctness:** 3
**Technical Novelty And Significance:** 4
**Empirical Novelty And Significance:** Not applicable
**Recommendation:** 5

**Clarity, Quality, Novelty And Reproducibility:**

The paper rates highly on each of clarity, quality and novelty. I only found minor typos in the writing.

**Strength And Weaknesses:**

Strengths.
1. Theoretical understanding of distribution shift is much needed, with the closest related work from 2010. Any progress toward better understanding is welcome.
2. Writing is mostly clear, assumptions are clearly stated and claims are argued well.

Weakness/questions.
1. I found the writing a bit dry because it does not anchor itself with interesting empirical findings that the theory can explain. They do not also have any section that explains how the theoretical results are of practical consequence.
2. I expected some empirical validation of their claims with either continuous or discrete inputs.
3. I find the considered class of functions (which are the sum of functions on each dimension) too restrictive. Does the bound on extrapolation measure $\tau$ depend on d because of the assumed class of models? How does the upper bound change with a more general class of models? For the continuous case, when does the bound on $\tau$ lead to a non-trivial bound? The paper can gain from some commentary around the developed theoretical claims, such as how tight the bounds are, what is the worst and best case shift, etc.

Minor comments.
1. Around eq (8), (9), r is swapped with k.
2. Section 3.3 talks about f1, f2 without introducing them.
3. “considered theorem 5” -> “considered in …”
4. Page 5: sparest cut -> sparsest cut.


**Summary Of The Paper:**

This paper studies extrapolation of models from source to target distribution. Unlike prior work that made restrictive assumptions about the distribution shift, this work considers any shift with shared marginals but arbitrary shifts in the joint. However, as a compromise, they only study a class of functions that are the sum of nonlinear functions on each coordinate (the last theorem relaxes the class of models to some extent though).

**Summary Of The Review:**

The paper makes a good effort toward developing theoretical bounds for extrapolation from a source to a target distribution for a specific class of models. The theoretical results are interesting but their relevance and impact value is left to the reader. The paper can improve by commenting on tightness of its result, discussing broader impact, and with engaging commentary in connection to known empirical results. Without such, it is difficult to assess the significance of its claims.

---

> ### Author Response · Authors · 2022-11-11
> **Response to Reviewer JEBk**
>
> We thank the reviewer for the comments. In the following, we will address the reviewer’s comments in detail.
>
> > I found the writing a bit dry because it does not anchor itself with interesting empirical findings that the theory can explain. They do not also have any section that explains how the theoretical results are of practical consequence.
>
> We kindly point out that the goal of this paper is not to explain empirical findings. Some level of extrapolation is observed for nonlinear models empirically, but the observation is not always reliable. Instead, the goal of this paper is to figure out any sufficient conditions for provable extrapolation.
>
> > I expected some empirical validation of their claims with either continuous or discrete inputs.
>
> We added some experiments on synthetic datasets accordingly, and we kindly refer the reviewer to the separate reply for our response to this question.
>
> > Does the bound on extrapolation measure $\tau$ depend on $d$ because of the assumed class of models? How does the upper bound change with a more general class of models?
>
> Indeed, our upper bounds on $\tau$ heavily depend on the structure of the models. The upper bound monotonically increases as we enlarge the model class. Eventually, $\tau$ converges to the density ratio if the model is an arbitrary function. Recall that even when the function class $\mathcal{F}$ is the family of two-layer neural networks, $\tau(P,Q,\mathcal{F})=\infty$ if the support of $Q$ is not a subset of the support of $P$ (Proposition 6). Therefore, any positive results of out-of-support extrapolation must restrict the function class so that it doesn’t include every two-layer neural network.
>
> > For the continuous case, when does the bound on $\tau$ lead to a non-trivial bound?
>
> As long as $\lambda(\Sigma_P)>0$, our upper bounds on $\tau$ are novel since the best-known result is the density ratio bound — when $\Sigma_Q\neq \Sigma_P$ the density ratio will be infinity (or exponentially large if the space is truncated). In contrast, our result $\tau<\infty$ implies that the OOD loss converges to zero if the ID loss converges to zero.

---

### Official Review · Reviewer_CdV3 · 2022-10-27

**Confidence:** 3
**Correctness:** 3
**Technical Novelty And Significance:** 2
**Empirical Novelty And Significance:** 3
**Recommendation:** 6

**Clarity, Quality, Novelty And Reproducibility:**

The results per se appear original to the best of my knowledge (proofs use rather standard ideas but nevertheless appear to be non-trivial). In terms of clarity, the paper would potentially benefit from better structuring. Despite some attempts of the authors to provide interpretations of the results more effort in this directions seems to be needed to make the contribution less "dry".

**Strength And Weaknesses:**

This work has some potentially interesting ideas, but overall the studied setting is limited making it questionable how informative or insightful the results really are.

The lower bound (Prop. 6), although not surprising, is helpful to the story as it motivates the idea of imposing structural assumptions between P and Q. Yet, the specific structural assumption assumed (identical marginals) is rather strong and at least it should be motivated. Besides that, the Gaussianity assumption also appears limiting in that setting. Even more concerning, there is no explicit discussion on how informative the bounds are (eg Thm 4 and 5). The result of Thm 4 is compared to the linear case but the comparison reads loose: (i) In (14) we have d/\lambda_min compared to 1/\lambda_min for linear models; why is "d" treated as a "constant" here? (ii) the bound in the linear case depends both on Q and P. (eg this is also the case in Thm, 3).

minor:
- Paper would benefit from a careful read as there are several typos here and there
- r_i not defined end of page 4. Why is wlog to assume i\in[2]
- Should it be f^* in (3)?

**Summary Of The Paper:**

The paper studies domain shift between distributions whose marginals have good overlap by analyzing the maximum of the ratio between the two distributions of the corresponding squared-distances between any two models. They bound this ratio for a model class that includes models additive over individual features and some additional gaussianity assumptions.

**Summary Of The Review:**

See above. I have not read the appendix due to review time constraints. At this point my feeling is that the paper is somewhat below the acceptance threshold. While the results could be of some interest, some more effort appears that is needed to justify the assumptions and draw concrete conclusions that are useful before the theory itself.


---------------------------------------------------
**** Raised score 5-->6. See response to authors

---

> ### Author Response · Authors · 2022-11-11
> **Response to Reviewer CdV3**
>
> We thank the reviewer for the comments, and for noting this work “has some potentially interesting ideas”. In the following, we will address the reviewer’s comments in detail.
>
> > Yet, the specific structural assumption assumed (identical marginals) is rather strong and at least it should be motivated.
>
> For the case with discrete features (Theorem 2), we only assume that the marginal distributions have a bounded density ratio. For the same reason, the bounded density ratio on the marginals is also sufficient for the setting of Theorem 4 & 5. However, bounded density ratio cannot be held for one-dimensional Gaussian distribution unless the marginals are identical. A possible (but not very clean) way to fix this is to work with truncated Gaussians with bounded density ratios on the marginals, which we leave as future work.
>
> > …the Gaussianity assumption also appears limiting in that setting.
>
> We kindly refer the reviewer to the separate reply for our response to this question.
>
> > Even more concerning, there is no explicit discussion on how informative the bounds are (eg Thm 4 and 5).
>
> On a high level, Theorem 4 and 5 prove the extrapolation for a richer set of nonlinear models with a coverage condition that is *comparable to linear models*. Since the best previous results are on linear models, our results are one of the first steps toward understanding nonlinear models’ extrapolation.
>
> > The result of Thm 4 is compared to the linear case but the comparison reads loose: (i) In (14) we have d/\lambda_min compared to 1/\lambda_min for linear models; why is "d" treated as a "constant" here? (ii) the bound in the linear case depends both on Q and P. (eg this is also the case in Thm, 3).
>
> We have a tighter bound for Theorem 4 that depends on both P and Q: $\tau \le \sup_{n\ge 1}\sup_{v\in\mathbb{R}^d}\frac{v^\top \Sigma_Q^{\odot n}v}{v^\top \Sigma_P^{\odot n}v}$​, where $\Sigma_Q^{\odot n}$ is the element-wise $n$-th power of the matrix $\Sigma_Q$ (please also see Eq. (56) in the appendix). This bound may be larger than its linear case counterpart (where we have $\tau\le \sup_{v\in\mathbb{R}^d}\frac{v^\top \Sigma_Qv}{v^\top \Sigma_Pv}$) because our model class is strictly larger. However, since we can prove that $\lambda_{\min}(\Sigma_P^{\odot n})\ge \lambda_{\min}(\Sigma_P)$ (Lemma 13), this tighter upper bound is at most $d/\lambda_{min}(\Sigma_P)$. We choose to present the current version of Theorem 4 for better exposition.
>
> The extra $d$ factor comes from the upper bound of the operator norm of $\Sigma_Q^{\odot n}$. Since we mostly focus on the conditions for source/target distributions (i.e., the $\lambda_{min}$ term), we omit the $d$ factor because it’s independent of $P$. We will update the paper accordingly to clearly state this difference.
>
> Re minor issues: we thank the reviewer for catching some typos. We will update the paper accordingly upon revision.
>
> > Why is wlog to assume i\in[2]
>
> For better exposition, we present the case where the number of features is 2 in Section 3.1. The general case is shown in Appendix A.5.
>
> > Should it be f^* in (3)?
>
> In (3) we should use $y(x)$ instead of $f^\star(x)$ because $y(x)$ represents the ground-truth labeling and $f^\star(x)\in\mathcal{F}$ is its best approximation in the function class.

---

### Official Review · Reviewer_fhp8 · 2022-10-28

**Confidence:** 2
**Clarity, Quality, Novelty And Reproducibility:** See previous section
**Correctness:** 4
**Technical Novelty And Significance:** 2
**Empirical Novelty And Significance:** Not applicable
**Recommendation:** 6

**Strength And Weaknesses:**

The most important addition to this paper would be a synthetic experiment where the data satisfies the assumption. This would both make it clearer how easy it is to construct such data and how tight the bounds are.

Discussing the impossibility results in previous work where disjoint support leads to unbounded divergence will also help with the motivation. Do Ben David et al already provide examples of this situation?

Minor comments:
- In Eq 1 tau is defined as a function of three variable and then, in the text below it, the variables are dropped. This change in notation should be clarified.
- I believe epsilon in Eq 2 needs a quantifier such as "there exists epsilon and f* such that...".
- What does stylized mean in the abstract?
- The title of the paper is a bit misleading. Ben David et al's result already applies to nonlinear models. The submission is providing results for certain classes of nonlinear models and data where the previous results would not apply and so is not "a first step" for the general case of nonlinear models.
- [1] also discusses issues with domain adaptation models and its relationship to support overlap. The discussion in [1] as far as I know is different from the submission but perhaps the authors can see some connections. There is no need to include this paper in related work if the authors do not find it relevant.

[1] Stojanov, Petar, et al. "Domain adaptation with invariant representation learning: What transformations to learn?." Advances in Neural Information Processing Systems (2021)

**Summary Of The Paper:**

This paper proposes domain generalization results for squared error that apply to cases where the support of the target distribution exceeds the support of the source distribution. There data model is nonlinear but the function has to be nearly realizable with a certain hypothesis class.

**Summary Of The Review:**

The approach taken in this paper is vastly different from previous domain adaptation works I've read and I lack the background to carefully evaluate correctness. I did my best to verify the math in the main paper but it's likely that I've missed some errors. The assumption on the data is restrictive but the paper is clear about it. I'm leaning towards acceptance.

---

> ### Author Response · Authors · 2022-11-11
> **Response to Reviewer fhp8**
>
> We thank the reviewer for the comments. In the following, we will address the reviewer’s concerns in detail.
>
> > The most important addition to this paper would be a synthetic experiment where the data satisfies the assumption.
>
> We kindly refer the reviewer to the separate reply for our response to this question.
>
> > Re: comparing with Ben-David et al. (“Do Ben David et al already provide examples of this situation?”), and the justification for the phase “first steps” in the title.
>
> On a high level, Ben-David et al *do not* provide any natural, concrete examples of nonlinear models and source and target domains where the model has good OOD performance. In contrast, we prove an upper bound for the OOD performance in *concrete settings* for nonlinear models, which is the first result of its kind to the best of our knowledge.
>
> We also note that our quantity $\tau$ is very similar to $H\Delta H$ divergence in Ben-David. We consider this kind of general quantities are almost sufficient and necessary (and thus a bit tautological) conditions for extrapolation to new domains.  On the other hand, our contribution is to bound $\tau$ for concrete nonlinear models and domains, and that are steps towards “understanding the extrapolation of nonlinear models”, e.g., when extrapolation can happen.
>
> That said, if the reviewer still believes that “first steps” overclaims, we are also happy to remove it from the title.
>
> > What does stylized mean in the abstract?
>
> By “stylized” we simply meant that our assumptions on the distribution shift and the model family are very much simplified and strong. However, we note that the best previous results are on linear models which are much less realistic, and our paper is one of the first steps toward understanding nonlinear models’ extrapolation.
>
> Re minor comments: we thank the reviewer for catching some typos and pointing out additional related works. We will update the paper accordingly upon revision.

---

### Author Response · Authors · 2022-11-11
**Common Questions and Answers**

We thank the reviewers for their constructive comments. On the positive side, the reviewers find this paper “has some potentially interesting ideas” (Reviewer CdV3), “makes a good effort toward developing theoretical bounds for extrapolation” (Reviewer JEBk), and “make a convincing case for why this result is powerful and novel” (Reviewer 1mvE).

The common major concerns of the reviewers are (i) we assume that each pair of features is pairwise gaussian distributed and it’s unclear whether this is true in practice (Reviewer CdV3, Reviewer 1mvE), and (2) whether our results are of practical relevance (Reviewer fhp8, Reviewer JEBk).

While the practical relevance of our theory would be a very interesting direction (and we thank the reviewers again for their constructive comments on this direction), the main goal of this paper is to figure out any sufficient conditions for provable extrapolation beyond the bounded density ratio case, which is an important theoretical question itself.

That said, in the following, we address these common concerns in detail.

> Justification of the pairwise Gaussianity of the features

On the theory side, our pairwise Gaussian assumption is substantially weaker than the joint Gaussian assumption required by some prior works on the theory of domain shift (e.g., [1,2]). Even with the Gaussianity assumption, existing results cannot prove any positive results for extrapolation to a distribution with different support.

On the empirical side, we can test whether the pairwise Gaussianity assumption holds approximately. On the iwildcam dataset, we take the output of the second-to-last layer of a Resnet50 pretrained with CLIP dataset as our features $x$, and we visualize the distributions of $(x_i, x_j)$ for randomly selected $(i,j)$ pairs in Figure 2 on Page 28 of the updated pdf. The pairwise Gaussianity assumption approximately holds in this case.

That said, we acknowledge that the Gaussianity assumption may not always hold in practice, and whether our results can be generalized to other distributions is left as future work.

[1] Chen, Yining, et al. Self-training avoids using spurious features under domain shift.
[2] Kumar, Ananya, et al. Understanding self-training for gradual domain adaptation.

> Empirical relevance of our results

On a very high level, our theory suggests that a structured model family can lead to substantially better extrapolation power (which we verify through synthetic datasets as shown below). Whether our theory inspires algorithms that can match/advance state-of-the-art performance is left to future works since this paper is mostly theoretical.

For the experiments on synthetic datasets, we focus on the setting of Theorem 5 where $x=(x_1,x_2)$ and $x_1\sim \mathbb{R}^{d_1},x_2\sim \mathbb{R}^{d_2}$. We compare the extrapolation of the structured model $f_1(x_1)+f_2(x_2)$ (where $f_1,f_2$ are two-layer neural networks) and unstructured model $f(x)$ (where $f$ is a two-layer neural network with input $[x_1,x_2]\in\mathbb{R}^{d_1+d_2}$). The ground-truth label is generated by a random structured model $f^\star_1(x_1)+f^\star_2(x_2)$. Recall that our theory shows that the structured model provably extrapolates to the target distribution. Indeed, empirically we found that the structured model has a significantly smaller *OOD* loss than the unstructured model even though the ID losses are comparable (Table  1, Page 26 of the updated pdf). Therefore, the structured model family has a *better extrapolation* than the unstructured family.

We refer the reviewers to Appendix D.1 in the updated manuscript for implementation details.

---

### Decision · Program_Chairs · 2023-01-20

**Decision:**

Accept: poster

**Justification For Why Not Higher Score:**

The paper is a borderline paper.

**Justification For Why Not Lower Score:**

See above

**Metareview: Summary, Strengths And Weaknesses:**

The paper poses an interesting problem to study: How do non-linear models extrapolate to new distributions. The author provide a precise definition for extrapolation to new distributions: Given a function class, they measure the ratio between the expected disagreement between any two functions in the class computed over the target distribution vs the expected disagreement computed over the source distribution. And the supremum over all the pairs of functions is used as a measure of extrapolation.

I have read the paper carefully myself, and discussed this with the reviewers. Overall, the problem formulation of the paper is very interesting, however the reviewers were not entirely convinced about some of the assumption (e.g. the additive function model analyzed or the Gaussianity) as well as whether/not there is a concrete message from the upper bounds (Thms 2-5) that non-linear models extrapolate better. I a revised version of the paper, the authors should clarify how useful these bounds are (it may be the case that they are too loose to be useful). Moreover, there is no comparison to lower bounds so tightness was also not clear to the reviewers.

Overall, the reviewers (and myself) see some good ideas in the work, but we recommend that the authors continue working on the results to achieve more concrete messages. These points were all mentioned in the reviews, and the authors took an extensive effort to respond (and agreed with the points). The paper is really on the border. As the AC, I would like to accept this paper because the paper provides new ideas and a problem formulation and the results should be discussed at the conference.



**Note From Pc:**

if the above contains the word "oral" or "spotlight" please see: "oral" presentation means -> notable-top-5% and "spotlight" means -> notable-top-25%. As stated in our emails, we are disassociating presentation type from AC recommendations

**Summary Of Ac-Reviewer Meeting:**

The main points are reflected in my meta-review. There was no clear preference to accept or reject. I vote for accepting the paper, because in general, I feel that the paper provides new ideas and a problem formulation, and it derives the chance to be be discussed at the conference.